# Snake Venom Hemotoxic Enzymes: Biochemical Comparison between *Crotalus* Species from Central Mexico

**DOI:** 10.3390/molecules24081489

**Published:** 2019-04-16

**Authors:** Octavio Roldán-Padrón, José Luis Castro-Guillén, José Alejandro García-Arredondo, Martha Sandra Cruz-Pérez, Luis Fernando Díaz-Peña, Carlos Saldaña, Alejandro Blanco-Labra, Teresa García-Gasca

**Affiliations:** 1Universidad Autónoma de Querétaro, Facultad de Ciencias Naturales, Avenida de las Ciencias S/N, Juriquilla, Querétaro 76230, México; octavio.roldan@uaq.mx (O.R.-P.); martha.sandra.cruz@uaq.mx (M.S.C.-P.); carlos.saldana@uaq.mx (C.S.); 2Centro de Investigación y de Estudios Avanzados del IPN, Unidad Irapuato, Irapuato, Guanajuato 36824, México; tekk.el.527@gmail.com; 3Universidad Autónoma de Querétaro, Facultad de Química, Cerro de las Campanas S/N, Centro Universitario, Querétaro 76010, México; alejandro.gr@uaq.mx (J.A.G.-A.); fer_diazp@hotmail.com (L.F.D.-P.)

**Keywords:** *Crotalus*, snake venom, phospholipases A_2_, hyaluronidases, metalloproteases, serine proteases

## Abstract

Snakebite envenoming is a serious medical problem in different areas of the world. In Latin America, the major prevalence is due to snakes of the family *Viperidae*, where rattlesnakes (*Crotalus*) are included. They produce hemotoxic venom which causes bleeding, tissue degradation and necrosis. Each venom has several enzymatic activities, producing different effects in the envenoming, doing its clinical effects difficult to study. Comparison between venom molecules is also difficult when different techniques are used, and therefore, their identification/characterization using the same methodology is necessary. In this work, a general biochemical characterization in snake venom of serine proteases (SVSP), phospholipases A_2_ (PLA_2_), metalloproteases (SVMP) and hyaluronidases (SVH) of *Crotalus aquilus* (Ca), *Crotalus polystictus* (Cp) and *Crotalus molossus nigrescens* (Cmn) was done. Differences in protein pattern, enzyme content and enzymatic activities were observed. All the venoms showed high PLA_2_ activity, high molecular weight SVSP, and a wide variety of SVMP and SVH forms. Ca and Cp showed the highest enzymatic activities of SVMP and SVSP trypsin-like and chymotrypsin-like, whereas Cmn showed the highest SVH and similar PLA_2_ activity with Ca. All the venoms showed peptides with similar molecular weight to crotamine-like myotoxins. No previous biochemical characterization of *C. aquilus* has been reported and there are no previous analyses that include these four protein families in these *Crotalus* venoms.

## 1. Introduction

Snakebite envenoming is a serious medical problem and yet some of the venom components have not been characterized. It is presently affecting people mainly from rural areas of tropical and subtropical regions of the world, mainly Asia, Africa, and Latin America. Annually, up to 5.4 million people are bitten by venomous snakes which cause between 20,000–94,000 deaths [1,2,3]. Snakes of the *Viperidae* family are responsible for a high number of bites in Latin America countries [4]. This family includes the *Crotalus* species (rattlesnakes) [2,5,6,7] that contain four main hemotoxic enzyme families: snake venom serine proteases (SVSP), snake venom hyaluronidases (SVH), snake venom metalloproteases (SVMP), and phospholipases A_2_ (PLA_2_) [8,9]. Although SVH have been reported, there is little information available [10,11]. The clinical effects of these type of venoms include inflammation, hemorrhaging at systemic and local levels, edema, tissue degradation, and necrosis [12]. The high variability in venom composition, which can produce different pathophysiological effects, as well as the synergistic and antagonistic effects of their components, make the complex events occurring during envenoming difficult to determine [13,14]. Therefore, the characterization and purification of venom components are key strategies in toxicology to understand the symptoms that occur during envenoming [9,15,16], however, only a few toxins have been completely examined [17]. In this work, venom from three *Crotalus* species was analyzed. *C. molossus nigrescens* (Cmn) is found in the Mexican highlands between Chihuahua and Oaxaca [18], *C. polystictus* (Cp) and *C. aquilus* (Ca) endemic species from central Mexico [19,20].

The enzymatic composition of *Crotalus molossus molossus* venom has already been studied, (serine proteases, metalloproteases and phospholipases), based mainly on the molecular weight determined by SDS-PAGE [21]. For Cmn, a thrombin-like serine protease was determined [22] and, by an analysis of the ontogenetic variation, the presence of crotamine, SVSP, SVMP, proteolytic, coagulant and fibrinogenolytic activities were reported by reverse phase high performance liquid chromatography (RP-HPLC) and western blot assays [23]. In the case of Cp, there is one characterization work by shotgun proteomic by LC-MS/MS and enzymatic activities determination for SVMP, SVSP, phosphodiesterase and L-amino acid oxidases (LAAO) [24]. However, for Ca only the protease activity has been identified [25] and there is not enough data about this venom composition. Presently, no previous biochemical characterization that includes the four enzyme families has been reported. Attention must be given to the high variability due to differences in the way samples are obtained, their preservation, type and concentration of substrates used to determine their activities, as well as the methodology used since these factors can influence the results [26,27]. In this work, we show novel data for the biochemical identification, enzyme activities and HPLC analysis for SVSP, PLA_2_, SVMP and SVH from the three snake *Crotalus* venoms using the same methodology in each case. The comparison between them can contribute to the development of specific snake bite treatments [21].

## 2. Results and Discussion

### 2.1. SDS-PAGE Protein Banding Pattern

Snake venoms of the *Viperidae* family consist of a complex mixture of proteins (up to 100 proteins), including different types of protein families [28]. The protein banding patterns of Ca, Cp and Cmn venoms under reducing and non-reducing conditions showed proteins with a broad range of molecular weights ranging from 6 to 180 kDa (Figure 1) with similarities in their protein pattern, but with some different protein bands, mainly in the range of 50 and 15 kDa. These differences have been observed in other snake venoms [14,29]. The protein patterns banding for Cmn and Cp are similar to previously reported data [23,30]. For Cp, LAAO, an abundant enzyme family related to the presence of yellow color in snake venoms [30,31] has been reported near 50 kDa [24]. In this work, all the venoms were yellow and showed a band of similar molecular weight that could correspond this enzyme (Figure 1), however, more studies and enzymatic activities determination for LAAO in these species, are still necessary to confirm its presence. For Cmn a 10 kDa crotamine-like myotoxin has been reported [23]. In this study, all the venoms showed protein bands in this molecular weight.

### 2.2. Zymography and Enzymatic Activities for Serine Proteases

Although SVSP has been reported in the range of 25–76 kDa [32], we detected bands in a broader range, between 15 and 95 kDa for trypsin-like (BApNA substrate) and 40 and 80 kDa for chymotrypsin-like (SAAFpNA substrate) serine proteases for Ca, Cp and Cmn venoms (Figure 2). These differences in molecular weight may be related to differences in their glycosylation patterns [33]. Chymotrypsin-like enzymes were less abundant than trypsin-like proteases, where BApNA substrate could also be recognized in general by serine proteases, such as thrombin-like and trypsin-like serine proteases [34]; these proteases are associated with the activation of metalloproteases zymogens [35]. Ca and Cmn venoms showed differences in SVSP trypsin-like in the range of 70 kDa, whereas for Cp venom did not appear to show activity in this molecular weight range. Cmn and Cp venoms also showed three proteases bands in a range of 14 to 25 kDa that were absent in Ca venom. For SVSP chymotrypsin-like proteases, Ca venom show two proteases bands, 40 and 80 kDa, whereas Cmn venom did not show the second one and in Cp venom both bands were absent. Some of the protein bands appear as weak bands in SDS-PAGE corresponding to trypsin-like proteases (Figure 1), which were charged with the same protein amount, this could be due to the difference in sensitivity of the technics, since zymography is a more sensitive method [36]. These protein bands can also be observed in the SDS-PAGE pattern banding and SDS-PAGE used for the chymotrypsin-like activity, which was done with a higher amount of venom protein (100 µg) (Figure 2).

Serine protease activity using BApNA substrate was higher than using SAAFpNA and elastase substrates in each case. Substrates with arginine in the P1 position, such as BApNA, can be hydrolyzed by trypsin-like and thrombin-like enzymes which are abundant components in snake venom [34]. Trypsin-like and chymotrypsin-like activities for Cp venom showed the highest values, while Ca and Cmn venoms showed the same activity. For elastase-like activity, Ca venom presented the lowest activity (Figure 3).

### 2.3. Zymography and Gelatinolytic/Caseinolytic Activities

In previous works, the proteolytic activity of SVMP was analyzed using the basement membrane’s protein degradation assay [16,37]. These enzymes can hydrolyze fibronectin, laminin, type IV collagen, and types I, III, and V gelatin [38]. The proteolytic activity of Ca, Cp, and Cmn venoms in gelatin zymography (Figure 4A) and the proteolytic activity using casein as substrate are shown (Figure 4B). Ca venom presented the highest caseinolytic activity followed by Cp and Cmn venoms. When the same activity was analyzed in the presence of EDTA, Ca and Cmn venoms showed no activity, while Cp venom kept up 40% activity, probably due to the presence of some serine proteases that recognize casein as substrate, as reported by Krogdahl and Holm (1983) for serine proteases of mammalians and birds [39]. These results are consistent with the highest trypsin and chymotrypsin-like serine protease activities showed by Cp venom. The decrease in all venoms activity by the chelating effect of EDTA indicates that the main proteolytic activity using casein as a substrate corresponds to metalloproteases. Gelatin zymography results showed that Ca venom presented the highest number of proteases, followed by Cmn and Cp venoms.

Since all SVMP activity are zinc-dependent [40], their presence was confirmed in gelatin zymography using EDTA (Figure 5). The SVMP of Ca, Cp and Cmn venoms corresponded to protein bands between 30 and 200 kDa, which were inactivated by EDTA at concentrations between 10 and 50 mM. The activity of most proteases in Cp venom was suppressed under these conditions, however, some proteases of MW of 40, 50, and 70 kDa remained active for Ca venom, and, other proteases of 70 and 90 kDa remained active for Cmn venom. These bands could correspond to thrombin-like enzymes since some of them can hydrolyze gelatin [41], however, the majority of serine proteases (detected by zymography) for Cp, and bellow 25 kDa for Ca and Cmn, showed no gelatinolytic activity.

Previous reports by Sánchez et al. (2001) and Chen and Rael (1997) showed the presence of two metalloproteases, one of 26 kDa and another of 25 kDa, respectively [42,43]. Ramírez (1990) reported the isolation of one protease and one thrombin-like serine protease in Cmn venom. In addition, for this species and for Cp venoms, Mackessy (2010) described the presence of SVMP with MWs of 21 and 45 kDa [21,22,44]. Our results show that Cmn venom has at least six SVMP bands of 30, 120, and 170 kDa and two of 200 kDa, indicating differences with the *C. molossus molossus* venom. Additionally, SVMP of 20, 50, 60, 75, 125 and 190 kDa bands were identified in Cp venom, with some differences in comparison with previous works that report SVMP of 20 and 50 kDa [24] by mass spectrometry. In the case of Ca venom, no previous reports for these proteases were found.

SVMP and SVSP have been reported as the main proteases for some *Viperidae* venoms [8,44,45]. For Ca, Cp, and Cmn venoms, most of the high molecular weight proteases corresponded to those two types of enzymes, probably due to the presence of homodimers or heterodimer oligomers, which have been reported for SVMP of the class P-III, PLA 2 and LAAO [46]. The identification of SVMP and SVSP (thrombin-like) in snake venoms is of particular interest since both activities have synergistic effects causing heavy bleeding [47,48]. Cp and Ca venoms show high activities of those type of proteases, similar and even higher than the activity for Cmn venom, which has been described as highly hemorrhagic and hemolytic [49,50]. For Cmn venom, at least two SVSP have been reported as the main compounds associated with the venom’s lethality [23].

### 2.4. Zymography and Enzymatic Activities of Phospholipases A_2_

Zymography of Cmn shows two protein bands (10 and 15 kDa), similar to the proteins previously reported by mass spectrometry in *C. molossus molossus* venom, 15.7 and 17 kDa [51] and 13.7 and 13.6 kDa [52]. For Ca and Cp venoms, 12 kDa and 13 kDa bands for PLA_2_ were found, respectively. These molecular weights were within the range reported for snake venom PLA_2_ (13–18 kDa) [53]. In this study, the protein bands for PLA_2_ for Ca, Cp, and Cmn venoms were different, however, the activity values of PLA_2_ for Ca and Cmn venoms were similar, whereas for Cp venom this enzymatic activity was the lowest. Compared with the positive control (bee PLA_2_), all the venoms showed higher enzymatic activity (Figure 6), which could be related to the myotoxic damage to muscle fibers caused by hemotoxic venoms [54]. For Cmn venom, the hemolytic effect has been related to the PLA_2_, associated with plasmatic membrane degradation [50].

### 2.5. Zymography and Enzymatic Activity of Hyaluronidases

The snake venom hyaluronidases (SVH) have been reported in a range from 33 to 110 kDa [55]. The highest variability was observed in the range of 20 to 50. However, Ramírez et al. (1990) reported that these enzymes were absent in Cmn venom [22] and for Cp venom, they have been detected by RP-HPLC/MS [30], whereas no previous reports for hyaluronidases were found for Ca venom. In this study, Ca, Cp, and Cmn venoms exhibited SVH with different MWs ranging from 10 to 150 kDa (Figure 7). Cp venom showed the highest activity (lower turbidity), followed by Ca and Cmn venoms. These activities were lower than the bovine testes hyaluronidase used as positive control. Snake venom hyaluronidases are known as dispersion factors, degrading hyaluronic acid of the extracellular matrix which can indirectly induce hemorrhaging and myotoxic damage related to SVMP and PLA_2_, respectively. They provoke a synergistic effect on tissue degradation and necrosis at the bite site; this component, however, has not been thoroughly studied [55,56].

### 2.6. RP-HPLC

Fractionation of venoms from the three species by RP–HPLC showed similar elution profiles, however, the concentrations among them were different. Abundant peaks ( ~10) between 40 and 55 min were found, where eluting times for proteins such as phospholipases A_2_ (35–45 min) [57], metalloproteases and hyaluronidases (≥50 min) [58], have been reported. Peaks appearing at 20 and 30 min were less abundant (~6), those peaks have similar eluting times to the previously reported for some serine proteases [59]. The last group of components, probably small peptides, eluted between 5 and 10 min [58]. Ca venom presented two peaks at 36 min, that were absent in Cp and Cmn venoms, while Cmn venom showed two peaks at 50 and 51 min that were not shared with the other two species. At 5 min, the most hydrophilic fractions were eluted in Ca and Cmn venoms. Cmn venom showed more variability, with more peaks eluting at 25, 47, and 50 min, followed by Ca and Cp venoms with less eluted fractions, respectively. The eluting retention times are consistent with reported data for phosfolipases A_2_, serine proteases (detected by zymography and enzymatic assays) and small peptides, however, the retention times for metalloproteinases and hylauronidases did not correspond with the retention times reported for other venoms, however, identification by enzymatic assays or MALDI-TOF Mass Spectrometry analysis are still needed, even though fractions were not detected after 60 min (Figure 8).

Cmn venom is considered highly hemorrhagic and hemotoxic [50,60]. Borja et al. (2018) reported that, using the Mexican antivenoms Antivipmyn^®^ and Faboterapico polivalente antiviperino^®^, this venom contains a large number of proteins with molecular weight below 20 kDa (included crotamine-like myotoxins), that were weakly or not recognized by western blot in juvenile individuals, and they also reported problems for neutralizing lethality of adult snake venoms [23]. Our results for Ca and Cp venoms showed similar patterns of proteins by SDS-PAGE, zymography and enzymatic activities with respect to Cmn venom, therefore it could be possible that recognition by the same antivenoms would also be similar. The molecular weight for SVSP, SVMP and SVPLA_2_ was similar to those observed by zymography.

## 3. Materials and Methods

### 3.1. Venom Extraction

The venom was obtained by manual extraction (milking) from four Ca, four Cp (adults of at least 45 cm length) and nine Cmn (adults of at least 60 cm length), with no differentiation by sex. The snakes were housed at the Autonomous University of Querétaro (Queretaro, Mexico) Animal Resource Facility, which was regulated under the General Wildlife Law. The protocol was approved by the Bioethics Committee of the Natural Sciences Faculty. Venoms were pooled, lyophilized, and stored at −70 °C until used. For all the analyses, the venom was dissolved in ddH_2_O and protein concentration in venom was determined [61] using BSA as a standard (Sigma-Aldrich, St. Louis, MO, USA).

### 3.2. SDS-PAGE Protein Banding Pattern

Twenty microgram of lyophilized venom protein was loaded into a 10 or 12% SDS-PAGE gel and electrophoresed in a Mini Protean II unit (Bio-Rad Hercules, CA, USA) or Hoefer *MiniVE system* (Hoefer, Holliston, MA, USA) under reducing (4% of β-mercaptoethanol and 5 min at boiling water temperature) and non-reducing conditions [62]. Seven microliter of unstained (Thermo Fisher Scientific, Waltham, MA, USA) or pre-stained (Sigma-Aldrich) MWM standards was used for molecular weight estimation. The gel was stained with 0.1% Coomassie brilliant blue R-250 (Bio-Rad) in 40% methanol and 10% acetic acid (*v/v*) overnight, de-stained in 40% methanol and 10% glacial acetic acid (*v/v*), and imaged on an HP Scanjet 4570c scanner.

### 3.3. Zymography Assays

#### 3.3.1. Serine Proteases Zymography

Venom samples were analyzed for SVSP, after electrophoresis, using 100 µg and 50 µg protein venom, for chymotrypsin-like and trypsin-like activities. A cellulose membrane (Bio-Rad) previously embedded with the specific substrate, for each serine protease was placed on top of the gel, incubated at 37 °C for 2.5 h (or until yellow bands in the gel were observed) and washed with 0.1% NaNO_3_, 0.5% (NH_4_)_2_SO_4_ in 1 M HCl and with 0.05% N-1-naphthylethylenediamine in 47.5% ethanol for periods of 5 min each. After the incubation time, the protein gel was stained/de-stained according to the method previously described. The substrates were prepared using 20 mM SAAPFpNA (N-Succinyl-Alanine-Alanine-Proline-Phenylalanine-p-nitroanilide, Sigma-Aldrich) for chymotrypsin-like enzymes and 20 mM BApNA (N-Succinyl-Arginine-p-nitroanilide, Sigma-Aldrich) for trypsin-like enzymes [63,64].

#### 3.3.2. Metalloproteases Zymography

Ten microgram of venom protein samples was run on polyacrylamide gels co-polymerized with 0.12% gelatin (Sigma-Aldrich) as a substrate [65]. The gel was washed in a 2.5% Triton X100 solution for 1 h and later for 30 min in a 0.05 M Tris-HCl (pH 7.4) buffer and finally incubated for 2–3 h in a 0.2 M NaCl, 0.005 M CaCl_2_, Triton X-100 0.002% (*v/v*), 0.001 M cysteine and 0.05 M Tris-HCl (pH 8) buffer. To confirm the presence of metalloproteases, venom samples were incubated for 30 min with EDTA (Bio-Rad) at 10, 50, and 100 mM concentrations in a buffer solution without CaCl_2_.

#### 3.3.3. PLA_2_ Zymography

After 20 µg of venom protein samples were run in a 10% SDS-PAGE gel under non-reducing conditions, the gel was washed for 1 h with Triton X-100 2.0% (*v/v*) and Tris-HCl 0.5 M, pH 7.4 buffer. The gel was subsequently washed for 30 min in a buffer with 140 mM NaCl, 2.5 mM CaCl_2_ and 50 mM Tris-HCl (pH 7.4) and then incubated for 2.5 h at 37 °C on a 1% agarose (Sigma-Aldrich) gel co-polymerized with a solution containing 50 mM Tris-HCl (pH 7.4), 140 mM NaCl, 2.5 mM CaCl_2_ and 2% egg yolk. The light zones indicated the presence of PLA_2_ [66].

#### 3.3.4. Hyaluronidases Zymography

A 10% SDS-PAGE gel was co-polymerized with hyaluronic acid (Sigma-Aldrich) and then 100 µg of venom protein were analyzed via non-reducing electrophoresis. The gel was washed 30 min with a 0.2 M sodium acetate (pH 6), 0.15 M NaCl and 2.5% (*v/v*) X-100 Triton buffer solution and subsequently incubated from 2.5 to 3 h at 37 °C in the same buffer solution without Triton. The gel was stained with alcian blue dye (Sigma-Aldrich) at 0.5% (*w/v*) and subsequently de-stained with a solution containing 40% methanol and 10% acetic acid [67].

### 3.4. Enzyme Assays

#### 3.4.1. Serine Protease Activities

The amidolytic activity was determined using 50 μg of venom protein, to measure trypsin-like, chymotrypsin-like, and elastase-like activities using BApNA, SAAPFpNA, and 10 mM N-Methoxysuccinyl-Ala-Ala-Pro-Val p-nitroanilide (Sigma-Aldrich) in dimethylsulfoxide as substrates, respectively. The mixture was brought to a final volume of 120 μL with 0.1 M Tris (pH 8) buffer and 10 μL of substrate was added to start the reaction. Activity was determined by the release of p-nitroanilide that absorbs at 405 nm, reporting as activity units (AU)/min/µg venom protein. One AU corresponded to an increase of absorbance of 0.01 at 405 nm [68]. Analyses were done in triplicate. In each case, bovine chymotrypsin, bovine trypsin and porcine elastase (Sigma-Aldrich) were used as positive controls.

#### 3.4.2. Proteolytic Activity

Proteolytic activity was assayed according to Gutiérrez et al. (2008) and Ponce et al. (2007) [69,70] with modifications. Briefly, 100 µg of venom protein and 500 µL of 2% (*w/v*) casein (Sigma-Aldrich) were diluted in 0.1 mM Tris–HCl (pH 7.4) and 0.15 M NaCl buffer. After 2.5 h of incubation at 37 °C, the reaction was stopped by adding 500 µL of 5% (*w/v*) trichloroacetic acid (TCA) at 37 °C for 30 min at room temperature. The mixture was centrifuged at 12,000 g for 15 min, and the casein hydrolysis was measured at 280 nm. Water was used as a negative control. Venom samples were incubated with 50 mM EDTA (Sigma-Aldrich) for 30 min before casein was added, to confirm the proteolytic activity of metalloproteases. The proteolytic activity was reported as activity units (AU)/µg venom protein where one activity unit corresponded to an increase of absorbance of 0.01 at 280 nm.

#### 3.4.3. Phospholipase A_2_ Activity

Phospholipase activity of 100 ng of venom protein was analyzed by a colorimetric assay using the Cayman Chemical kit (Ann Arbor, MI, USA, catalogue number 765001). The assay used 1,2-dithio diheptanoyl phosphatidylcholine analogue as substrate. The free thiols generated by hydrolysis of the thioester bonds were detected using DTNB [(5,5′-dithio-bis- (2-nitrobenzoic acid)]. The increase in absorbance was measured in a spectrophotometer (Benchmark Plus, Bio-Rad, USA) at 414 nm every minute for 10 min. PLA_2_ of bee venom was used as a positive control. The enzymatic activity was expressed in substrate μmol hydrolyzed/min/μg protein.

#### 3.4.4. Hyaluronidase Activity

Venom protein concentrations ranging from 20 to 200 μg were mixed with 100 μL of 0.2 M sodium acetate (pH 6), 0.15 M NaCl buffer, and 100 μL of 1 mg/mL solution of *Streptococcus equi* hyaluronic acid (Sigma-Aldrich) as a substrate for a final volume of 250 μL. After 15 min incubation at 37 °C, the reaction was stopped by adding 1 mL of 2.5% hexadecyl trimethyl ammonium bromide in 2% NaOH for 30 min at room temperature. Turbidity was determined at 400 nm in a plate reader using a spectrophotometer (Benchmark Plus, Bio-Rad, USA). Bovine testes type IV-S hyaluronidase (Sigma-Aldrich) was used as control [71].

### 3.5. RP-HPLC

For RP-HPLC analysis, 200 μg of venom protein from pooled venoms of Ca, Cp and Cmn were used, by using a C18 symmetry column (150 × 4.6 mm. Particle size: 3.5 μm) equilibrated with solution A (H_2_O at 0.1% with TFA), using an Agilent 1200 chromatograph. Elution was performed at 1 mL/min using a gradient towards solution B (CH_3_CN at 0.1% with TFA) as follows: 0% B by 5 min, 0%–60% B over 60 min. Absorbance was monitored at 280 nm.

### 3.6. Statistical Analysis

To evaluate differences between venom enzymatic activities, data were analyzed using the *GraphPad Prism V5.0* software by ANOVA (Tukey *p* < 0.05).

## 4. Conclusions

Analysis of three different *Crotalus* species: *Crotalus aquilus* (Ca), *Crotalus polystictus* (Cp) and *Crotalus molossus nigrescens* (Cmn) *Crotalus* snake venoms showed differences between the quality and quantity of the studied enzymes. Some of the proteases found in the studied species showed molecular weights not previously reported that also differed among the analyzed venoms, pointing to the importance of studying the wide diversity composition between *Crotalus* species. Some of these differences of the observed bands could correspond to oligomeric forms of enzymes previously reported present in other snake venoms, which could also explain the absence in the zymography of the 25 kDa SVMP reported. Our results show that Ca, Cp and Cmn venoms are mainly hemotoxic with the main groups of enzyme families usually found in *Viperidae* venom: metalloproteases, serine proteases, phospholipases A2, and hyaluronidases, in high quantities and a variable number of peptides, probably crotamine-like myotoxins. These enzymes are present in different proportions and different enzymatic activities, depending on the species. Ca venom, showed the highest SVMP activity, with a greater number of bands of high molecular weight, whereas Cp venom showed the highest activities for SVSP (trypsin and chymotrypsin-like) and SVH proteases; Cmn venom presented similar activity level for SVSP activity (elastase-like) and just higher activity of PLA2 than Cp venom, but with less abundance of high molecular weight SVMP. The PLA2 enzymatic activity was highly active in all the venoms. In Cmn, SVSP activity, and also some small proteins and peptides (lower than 20 kDa) with low recognition by two antivenoms, plays an important role in venom toxicity. The enzymatic activities as by zymography and RP-HPLC data showed similar patterns between venoms. We suggest that Ca and Cp venoms could have a similar high hemorrhagic, hemotoxic and hemolytic effects as has been reported for Cmn. Finally, the presence of a wide range of hydrolytic enzymes in the studied venoms may play an important role in their toxic effect. Knowledge of venom variability could be useful in developing more effective strategies in anti-venom therapy. Besides, characterization of the different proteases could also be of importance for their potential biotechnological application. Further work will determine the molecular identity of the main enzymes for each snake venom in order to have a better knowledge of therapeutic antidotes.

## Figures and Tables

**Figure 1 molecules-24-01489-f001:**
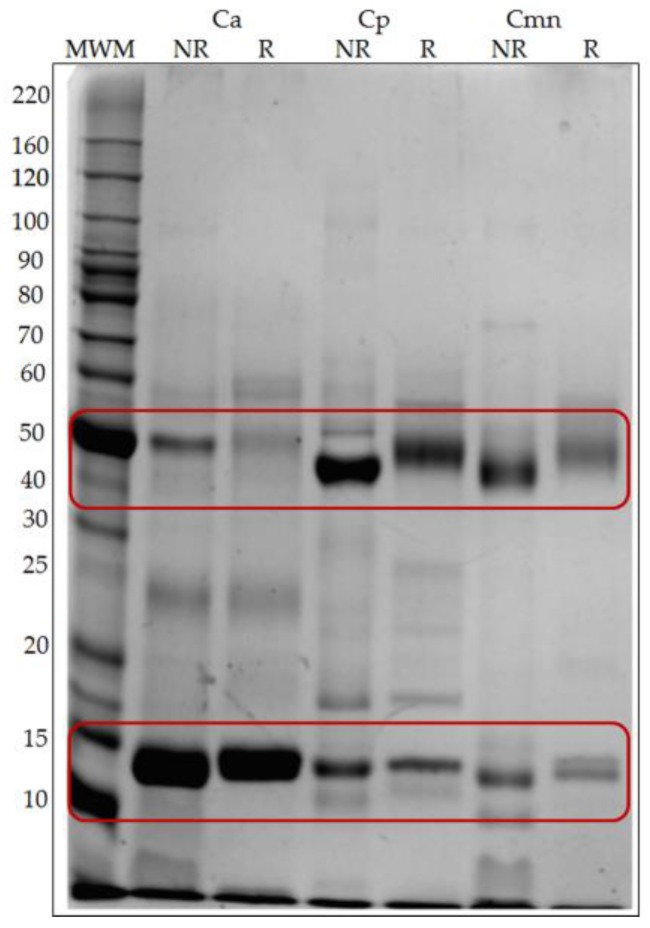
Sodium dodecyl sulfate polyacrylamide gel electrophoresis (SDS-PAGE) protein banding pattern for *C. aquilus* (Ca), *C. polystictus* (Cp) and *C. molossus nigrescens* (Cmn) venoms. Twenty micrograms of lyophilized venoms were analyzed by non-reducing (NR) and reducing conditions (R) in 10% polyacrylamide gels. MWM: molecular weight markers (kDa). Variation in the bands with molecular weight ~15 kDa and ~50 kDa are marked with red boxes.

**Figure 2 molecules-24-01489-f002:**
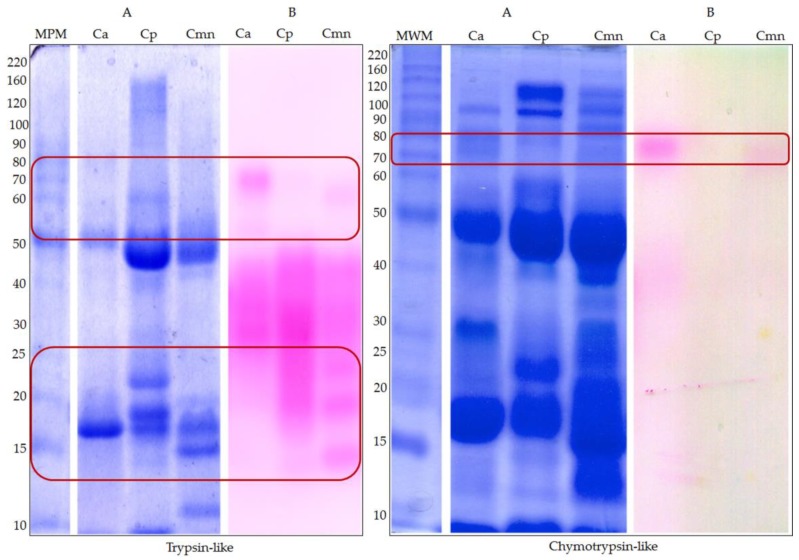
Protein pattern by SDS-PAGE, and Serine proteases zymography for *C. aquilus* (Ca), *C. polystictus* (Cp) and *C. molossus nigrescens* (Cmn) venoms. (**A**) 10% SDS-PAGE and (**B**) zymography. Trypsin-like zymography was performed using 50 μg of venom samples protein with BApNA substrate. Chymotrypsin-like zymography was determined using 100 μg of venom protein with SAAFpNA substrate. After electrophoresis, a cellulose membrane with each serine protease was placed on top of the gel and incubated for 5 h and 2.5 h at 37 °C, respectively. MWM: molecular weight markers (kDa). Variability in proteases bands is marked with red boxes.

**Figure 3 molecules-24-01489-f003:**
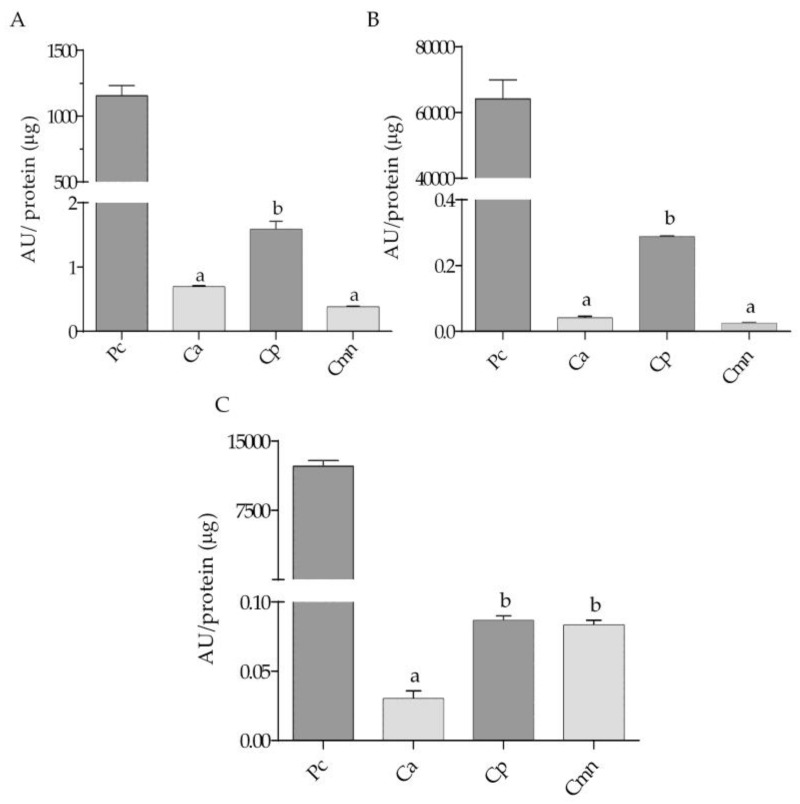
Snake venom serine proteases activities for *C. aquilus* (Ca), *C. polystictus* (Cp) and *C. molossus nigrescens* (Cmn) venoms. (**A**) Trypsin-like proteases, using bovine trypsin as positive control; (**B**) chymotrypsin-like proteases, using bovine chymotrypsin as positive control; (**C**) elastase-like proteases, using porcine elastase as a positive control. Enzymatic activity expressed in activity units (AU/µg). Small letters show significant differences (Tukey *p* < 0.05) between samples for each enzyme, positive controls (Pc) were not compared.

**Figure 4 molecules-24-01489-f004:**
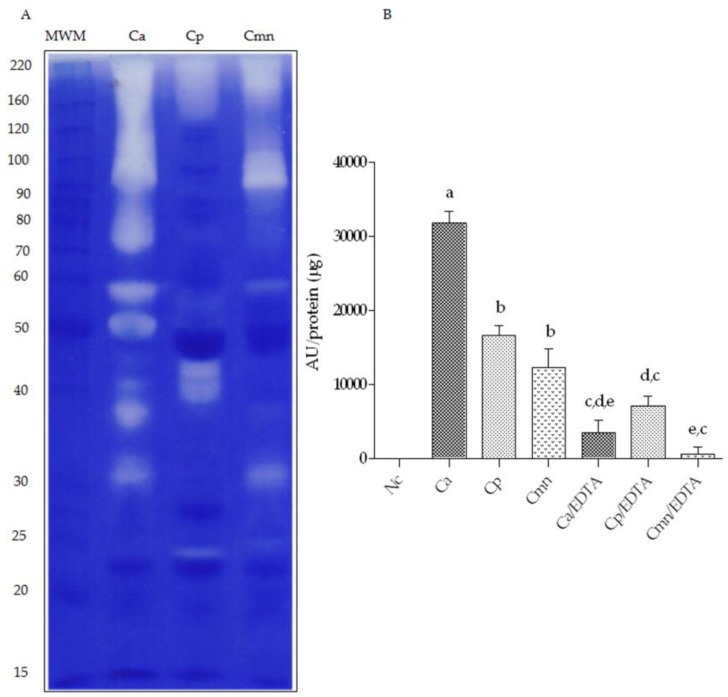
Gelatin zymography in 10% polyacrylamide gels and proteolytic activity using casein substrate. (**A**) Ten micrograms of venom protein from *C. aquilus* (Ca), *Crotalus polystictus* (Cp) and *C. molossus nigrescens* (Cmn) were incubated after electrophoresis for 2.5 h at 37 °C. MWM—molecular weight markers (kDa). (**B**) Proteolytic activity of 100 μg of venom samples in the presence/absence of 50 mM EDTA incubated with casein as a substrate for 2.5 h at 37 °C. Negative control (Nc). Small letters (a,b,c,d,e) show statistical difference (Tukey *p* < 0.05). Variability in proteases bands is marked with red boxes. Small letters show significant differences (Tukey *p* < 0.05) between samples, either with or without ethylendiaminetetraacetic acid (EDTA).

**Figure 5 molecules-24-01489-f005:**
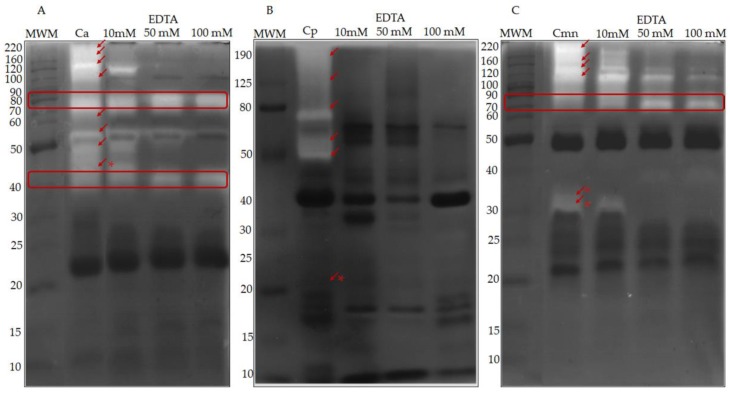
Zymography in 10% polyacrylamide gels co-polymerized with gelatin. Ten microgram of venom protein from (**A**) *C. aquilus* (Ca), (**B**) *C. polystictus* (Cp), and (**C**) *C. molossus nigrescens* (Cmn). In addition, 10 μg of the same samples were tested with 10, 50 and 100 mM EDTA in gels that were incubated after electrophoresis for 2.5 h at 37 °C. Red arrows indicate inhibition of snake venom metalloproteases (SVMP) by EDTA, red asterisk (*) show bands not shared between the venoms, red boxes show proteases bands with gelatinolytic activity unaffected by EDTA. Molecular weight markers (MWM) in kDa.

**Figure 6 molecules-24-01489-f006:**
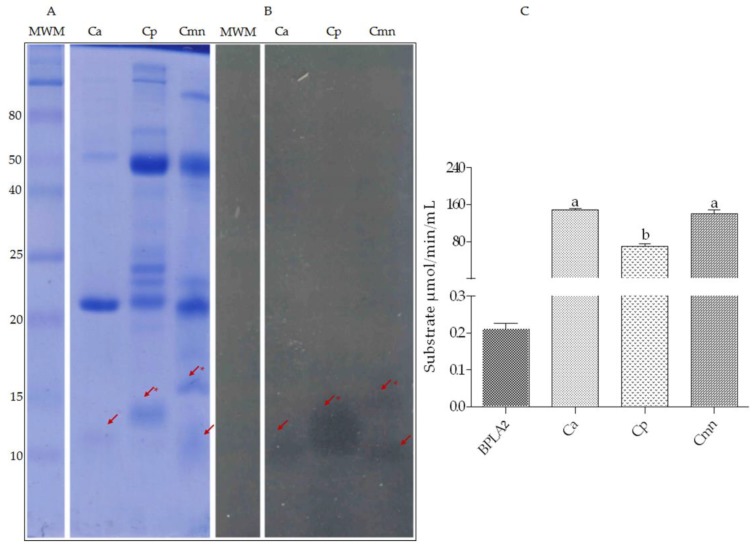
Zymography and enzymatic activity of PLA2. (**A**) 12% SDS-PAGE, (**B**) 4% agarose-egg yolk co-polymerized gel zymography using 20 μg of protein from *C*. *aquilus* (Ca), *C. polystictus* (Cp), and *C*. *molossus nigrescens* (Cmn) venoms incubated after electrophoresis, for 2 h at 37 °C. MWM—molecular weight markers (MW) in kDa. (**C**) Enzymatic activity of 10 μg of bee venom phospholipase (BPLA2) as a positive control and 100 ng from Ca, Cp and Cmn venoms. Small letters (a,b) show statistical differences (Tukey *p* < 0.05) between samples, positive control was not compared. Red arrows show bands with PLA_2_ activity, and red asterisk show bands not shared between the venoms.

**Figure 7 molecules-24-01489-f007:**
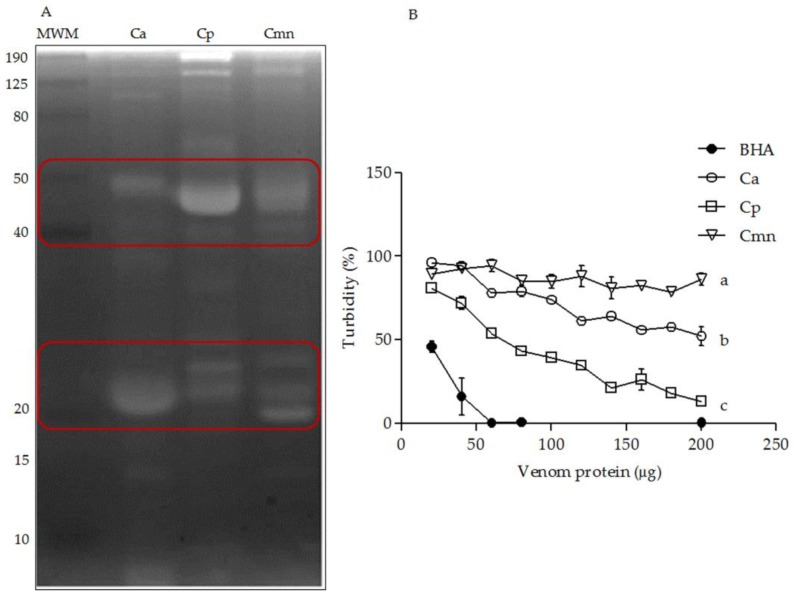
Hyaluronidase activity. (**A**) Zymography on SDS-PAGE co-polymerized with hyaluronic acid using 100 μg venom protein, incubated after electrophoresis for 2.5 h at 37 °C. Molecular weight markers (MWM) in kDa. (**B**) Enzymatic activity using 5 to 50 μg of *C. aquilus* (Ca), *C. polystictus* (Cp), and *C. molossus nigrescens* (Cmn) venoms as well as hyaluronidase from bovine testes used as positive control (BHA). Red boxes show variability between the venoms. Small letters show statistical differences (Tukey *p* < 0.05) between samples, positive control was not compared.

**Figure 8 molecules-24-01489-f008:**
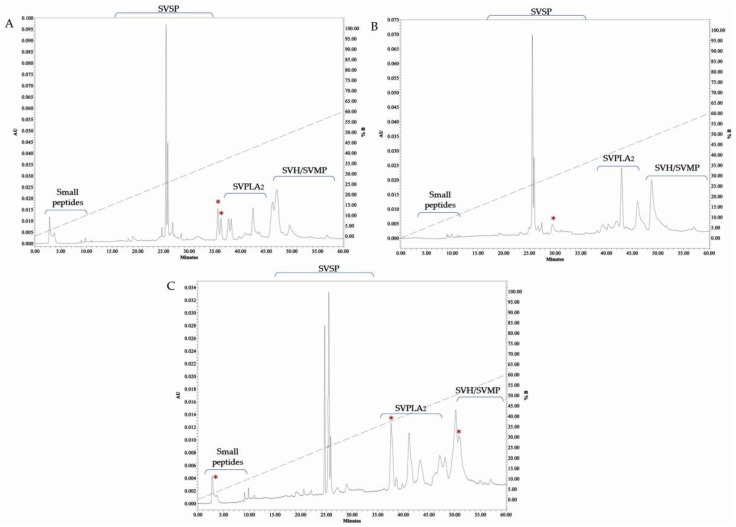
Venoms fractionation by RP–HPLC. Pooled *crotalus* venoms from (**A**) *C. aquilus* (Ca), (**B**) *C. polystictus* (Cp) and (**C**) *C. molossus nigrescens* (Cmn). Red asterisk show peaks not shared between the venoms. Snake venom serine proteases (SVSP), snake venom metalloproteases, (SVMP) and snake venom hyaluronidases (SVH). Absorbance units (AU) at 280 nm. Enzymatic activities were not determined, the family proposed for the enzymes are based on similarity of the elution profiles with reported data.

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
