# Peer review of "Snake Venom Hemotoxic Enzymes: Biochemical Comparison between Crotalus Species from Central Mexico"

_molecules, 2019, doi:10.3390/molecules24081489_

Round 1
Reviewer 1 Report
This manuscript reports a biochemical comparison of snake venom hemotoxic enzymes from central Mexican Crotalus species.
The authors collected and pooled venom from three species of Crotalus from central Mexico. The venom from Crotalus polystictus (Cp), C. aquilus (Ca) and C. molossus nigrescens (Cmn) was tested and analyzed using a number of different techniques including SDS-PAGE, RP-HPLC and MALDI-TOF MS for compositional analysis, and zymography and enzymatic assays for activity analysis. The enzyme activities studied on the four primary hemotoxic enzyme families found in Crotalus snake venoms; serine proteases (SVSP), metalloproteases (SVMP), hyaluronidases (SVH) and phospholipase A2 (PLA2). The SDS-PAGE results showed similar banding patterns for the three different venoms, however differences were also evident. Likewise, similar RP-HPLC chromatograms were obtained but with observable differences in peak presence and abundance. The MALDI-TOF MS data showed molecular weight peaks consistent with the SVSP, SVMP and PLA2 proteins, and lower molecular weight peaks (6.7-11 kDa), predicted to be crotamine-like myotoxins, in the three samples. The enzyme activity studies showed Ca venom had the highest SVMP activity, while Cp had the highest SVSP and SVH activities. Cmn had similar levels of SVSP (elastase-like) activity and higher PLA2 than Cp.
The main conclusions reported for the work are that C. polystictus, C. aquilus and C. molossus nigrescens venoms are hemotoxic and contain the four main protease families found in Viperidae venom. Additionally, Crotalus polystictus and C. aquilus venom could have similar activity to the previously studied C. molossus nigrescens venom.
Overall the work is interesting. The manuscript requires English copyediting to help clarify some of the points of the study.
General comments:
The RP-HPLC and MALDI-TOF MS peaks are labelled based on the predicted protein family they correspond to. These peaks could have been easily further confirmed by fractionating the RP-HPLC peaks and performing MALDI-TOF MS of the collected fractions to provide specific mass data with peaks present in the RP-HPLC.
What are the age and sex of the snakes? Ontonogenic differences are mentioned, but is there also any known sex related differences in the venom?
Page 2, line 57: The sentence beginning “For Cmn…” is not clear and requires rewording.
Figure 3, Page 4: Panel D is the negative controls, so the y-axis label of venom protein is not correct.
Page 7, line 212: “Abundant peaks between 40 and 55 were found” requires units.
Page 8, line 214: The sentence beginning “Less peaks…” is not clear and requires rewording.
Page 8, line 219: The sentence beginning “The fractionation…” is not clear and requires rewording.
Page 14, Ref 19: Insufficient reference detail.
Page 15, Ref 41: Is ref 41 a duplicate of reference 21?
There are a significant number of typographical and grammatical errors, and the manuscript requires English copyediting. A limited number of examples are listed below, but this list is far from comprehensive.
Typographical and grammatical errors:
Page 1, line 31: “miotoxins” should read “myotoxins”.
Page 2, line 46: “hemorrhages” should read “hemorrhaging”.
Page 2, line 52: The sentence beginning “In this work…” should read “In this work, venom from three Crotalus species was analyzed.”.
Page 2, line 57: “their molecular” should read “the molecular”.
Page 2, line 66: “obtainment” should read “obtained”.
Page 2, line 77: “type” should read “types”.
Page 2, line 78: “range molecular” should read “range of molecular”.
Page 5, line 150: “thrombine” should read “thrombin”.
Page 7, line 198: “than bovine testes hyaluronidase used as positive control” should read “than the bovine testes hyaluronidase used as a positive control”.
Page 7, line 200: “hemorrhages and myotoxic damage related with” should read “hemorrhaging and myotoxic damage related to”.
Page 8, line 211: “species, by” should read “species by”.
Page 8, line 217: “absents in Cp” should read “absent in the Cp”.
Page 8, line 217: “2 peaks” should read “two peaks”.
Page 8, line 219: “peaks eluted” should read “peaks eluting”.
Page 9, line 231: “about of” should read “approximately”.
Page 9, lines 232,33: Bracketing is incorrect and requires correction.
Page 9, line 236: “6,7 kDa” should read “6.7 kDa”.
Page 9, line 239: An extra space is present between “case of”.
Page 10, line 256: “possess” should read “possesses”.
Page 10, line 266: “not share peak” should read “peaks not shared”.
Page 11, line 266: “were” should read “was”.
Page 11, line 308: “one h” should read “1 h”.
Page 12, line 358: “poled” should read “pooled”.
Page 7, lines 260, 261: “Figure 2A” should read “Figure 3A”, and “Figure 2B” should read “Figure 3B”.
Author Response
Reviewer 1
The RP-HPLC and MALDI-TOF MS peaks are labelled based on the predicted protein family they correspond to. These peaks could have been easily further confirmed by fractionating the RP-HPLC peaks and performing MALDI-TOF MS of the collected fractions to provide specific mass data with peaks present in the RP-HPLC.
Due to the comments of one of the reviewers, the MALDI-TOF MS analysis was eliminated. We agree with your observation and the peaks will be confirmed in the future by further fractionation. Thank you.
What are the age and sex of the snakes? Ontonogenic differences are mentioned, but is there also any known sex related differences in the venom?
The venom was extracted from adult snakes: 45 cm length C. aquilus/C. polystictusadults and 60 cm C. molossus nigrecsens. without distinction of sex due to the number of individuals (page 10, lines 265 and 266).
Page 2, line 57: The sentence beginning “For Cmn…” is not clear and requires rewording.
The correction was done (page 2, lines 59-61).
Figure 3, Page 4: Panel D is the negative controls, so the y-axis label of venom protein is not correct.
The correction was done (page 5, line 127).
Page 7, line 212: “Abundant peaks between 40 and 55 were found” requires units.
Units were added (page 9, line 232).
Page 8, line 214: The sentence beginning “Less peaks…” is not clear and requires rewording.
The correction was done (page 9, lines 234-236).
Page 8, line 219: The sentence beginning “The fractionation…” is not clear and requires rewording.
The correction was done (page 9, lines 239-241).
Page 14, Ref 19: Insufficient reference detail.
The reference 19 was corrected (page 14).
Page 15, Ref 41: Is ref 41 a duplicate of reference 21?
The reference 41 was deleted due that was a duplicate of reference 21 (page 14).
There are a significant number of typographical and grammatical errors, and the manuscript requires English copyediting. A limited number of examples are listed below, but this list is far from comprehensive.
Typographical and grammatical errors:
Page 1, line 31: “miotoxins” should read “myotoxins”.
The correction was done (page 1, line 32).
Page 2, line 46: “hemorrhages” should read “hemorrhaging”.
The correction was done (page 2, line 48).
Page 2, line 52: The sentence beginning “In this work…” should read “In this work, venom from three Crotalus species was analyzed.”
The correction was done (page 2, lines 53-54).
Page 2, line 57: “their molecular” should read “the molecular”.
The correction was done (page 2, line 58).
Page 2, line 66: “obtainment” should read “obtained”.
The correction was done (page 2, line 67).
Page 2, line 77: “type” should read “types”.
The correction was done (page 2, line 76).
Page 2, line 78: “range molecular” should read “range of molecular”.
The correction was done (page 2, line 79).
Page 5, line 150: “thrombine” should read “thrombin”.
The correction was done (page 6, line 161).
Page 7, line 198: “than bovine testes hyaluronidase used as positive control” should read “than the bovine testes hyaluronidase used as a positive control”.
The correction was done (page 8, lines 217-218).
Page 7, line 200: “hemorrhages and myotoxic damage related with” should read “hemorrhaging and myotoxic damage related to”.
The correction was done (page 8, lines 219-220).
Page 8, line 211: “species, by” should read “species by”.
The correction was done (page 9, line 231).
Page 8, line 217: “absents in Cp” should read “absent in the Cp”.
The correction was done (page 9, lines 237).
Page 8, line 217: “2 peaks” should read “two peaks”.
The correction was done (page 9, line 237).
Page 8, line 219: “peaks eluted” should read “peaks eluting”.
The correction was done (page 9, line 240).
Page 9, line 231: “about of” should read “approximately”.
The MALDI-TOF-MS was deleted due to the comments of one of the reviewers. This assay will be considered for further analyses.
Page 9, lines 232,33: Bracketing is incorrect and requires correction.
The MALDI-TOF-MS was deleted due to the comments of one of the reviewers. This assay will be considered for further analyses.
Page 9, line 236: “6,7 kDa” should read “6.7 kDa”.
The MALDI-TOF-MS was deleted due to the comments of one of the reviewers. This assay will be considered for further analyses.
Page 9, line 239: An extra space is present between “case of”.
The MALDI-TOF-MS was deleted due to the comments of one of the reviewers. This assay will be considered for further analyses.
Page 10, line 256: “possess” should read “possesses”.
The MALDI-TOF-MS was deleted due to the comments of one of the reviewers. This assay will be considered for further analyses.
Page 10, line 266: “not share peak” should read “peaks not shared”.
The MALDI-TOF-MS was deleted due to the comments of one of the reviewers. This assay will be considered for further analyses.
Page 11, line 266: “were” should read “was”.
The correction was done (page 10, line 261).
Page 11, line 308: “one h” should read “1 h”.
The correction was done (page 11, line 303).
Page 12, line 358: “poled” should read “pooled”.
The correction was done (page 12, line 355).
Page 7, lines 260, 261: “Figure 2A” should read “Figure 3A”, and “Figure 2B” should read “Figure 3B”.
The figures are correctly marked and correctly numbered (page 3, line 106).

Reviewer 2 Report
Abstract: „Each enzyme can have more than one enzymatic activity…” I cannot agree with this generalization. What did authors have in mind? It is true that there exist bi-functional enzymes, as well as it is known that enzymes in subcritical conditions can catalyze reverse reaction but hydrolases are in general able to hydrolyze more or less specific type of bond as well as transferases are responsible for transfer of specific functional groups. What do Authors mean by “more than one enzymatic activity” ?
2.1 SDS-PAGE protein banding pattern
Electrophoresis should be internally standardized with protein concentration in venom not concentration of venom itself. There is a very little number of protein bands in your gels. In Viperidae venoms there are probably over a hundred proteins, so why is there only a dozen bands in your gels?
2.2 Zymography and enzymatic activities for serine proteases
In my opinion presented results are inconclusive. Bands on the zymogram especially for trypsin-like proteases are blurred beside the fact that lower box marking specific areas is too small. I suggest to repeat this experiment with lower venom concentration.
2.3 Metalloproteases Zymography
Caption below figure 4 is incomplete or even confusing: “A) Ten micrograms of venom protein from C. aquilus (Ca), Crotalus polystictus (Cp) and C. molossus nigrescens (Cmn) were incubated for 2.5 h at 37° C”. Do you mean venom before electrophoresis, gel after electrophoresis or something else? Besides, obtained lines on zymogram have more differences than similarities. There is no point to indicate some specific differences where there is so many of them.
How do you explain the lack of presence of SVMP described in literature (about 25 kDa)? Is it possible that you don’t see monomeric forms of SVMP in the conditions you run the gels?
According to the obtained results that clearly indicate that not only metaloproteases were separated on gels with gelatin, I suggest the change of paragraph title. Metalloproteases Zymography suggests that SVSPs have not gelatinolytic activity and all observed results concern SVMPs. These results are very interesting and I do not deny their value but description in this case is a bit confusing.
Line 166 “In snake venoms, the most abundant components are SVMP and SVSP” this is non generally true. It may be for some viperid venoms, but there are also a lot of PLA2s. But it is absolutely not true for elapid venoms, where 3FTx toxins are the principal component.
2.6 RP-HPLC
On the figure 8 (in the caption of course) it should be clearly marked that proteins in presented peaks were not identified. They are predicted on the base of literature data and there is no proof that mentioned proteins are actually present in those peaks.
2.7 MALDI-TOF-MS
In my opinion this experiment does not have scientific value. There is no identification data for proteins in particular peaks. All information is a prediction on the base of literature data and from proteomic point of view they are illegitimate. How do you know that there are no multiple-charged ions on this mass spectrum? You reported that in analyzed venoms you observed proteins with higher mass than 50 kDa. Presented mass spectrum “ends” with 50 000 m/z (by the way it is not necessarily equal to mass) and most of enzymes analyzed in this paper have higher mass. Besides, while the mass of PLA2 observed on zymography agrees with m / z on mass spectra, in the case of SVMP the difference is huge. The zymography with gelatin indicate SVMPs with the mass of 80 kDa and more not about 25. In both paragraphs (2.3 and 2.7) you write that there are reports about SVMPs with lower mass than you observed. How can you explain this difference? Concluding, the mere comparison of mass spectra is interesting but it does not add anything. In my opinion these result does not have value and conclusions are not supported by facts.
3. Materials and Methods
Nowhere is there any information on how the venom was dissolved either for electrophoresis or for enzyme tests.
4. Conclusions
“Some of the proteases found in the studied species showed molecular weights not previously reported…” these differences probably result from the fact that you observe multimeric forms of proteases. This subject should be discussed in this article.
“Our results show that Ca, Cp and Cmn venoms are mainly hemotoxic with the main groups of protease families usually found in Viperidae venom: metalloproteases, serine proteases, phospholipases A2, and hyaluronidases…” - neither phospholipases nor hyaluronidases are peptidases.
Author Response
Reviewer 2
Abstract: „Each enzyme can have more than one enzymatic activity…” I cannot agree with this generalization. What did authors have in mind? It is true that there exist bi-functional enzymes, as well as it is known that enzymes in subcritical conditions can catalyze reverse reaction but hydrolases are in general able to hydrolyze more or less specific type of bond as well as transferases are responsible for transfer of specific functional groups. What do Authors mean by “more than one enzymatic activity” ?
We have done the suggested change: “Each enzyme can have more than one enzymatic activity” for “Each venom has several enzymatic activities”… (page 1, line 21).
2.1 SDS-PAGE protein banding pattern
Electrophoresis should be internally standardized with protein concentration in venom not concentration of venom itself.
According to the methodology, protein in venom was determined by Bradford assay. Electrophoresis was done base on protein concentration (page 10, lines 270-272).
There is a very little number of protein bands in your gels. In Viperidae venoms there are probably over a hundred proteins, so why is there only a dozen bands in your gels?
There are approximately 12 protein bands in SDS-PAGE gels; however each protein band could contain more than one protein, also, the protein banding pattern of Cmn and Cp were similar to the results reported by Borja et al. (2018) [23]and Mackessy et al. (2018) (page 2, lines 80-81).
2.2 Zymography and enzymatic activities for serine proteases
In my opinion presented results are inconclusive. Bands on the zymogram especially for trypsin-like proteases are blurred beside the fact that lower box marking specific areas is too small. I suggest to repeat this experiment with lower venom concentration.
The paragraph explaining the enzymatic activities of serine proteases was reworded and box indicating differences between the venoms was completed (page 3, lines 101-106). Also, the experiment using lower venom concentrations was repeated, however,the zymography resulted blurred and some trypsin like serine proteases of lower molecular mass were not detected, for this reason we kept the original Figure 2 in the manuscript (page 4, line 113).
Protein pattern by SDS-PAGE, and Serine proteases zymography for C. aquilus(Ca), C. polystictus(Cp) and C. molossus nigrescens (Cmn) venoms. (A) 10% SDS-PAGE and (B) Zymography. Trypsin-like zymography was performed using 50 μg (left) and 25 μg (right) of venom protein samples with BApNA substrate. After electrophoresis, according to methodology, a cellulose membrane with each serine protease was placed on top the gel and incubating for 2.5 h at 37 °C MWM Molecular weight markers (kDa). Proteases of low molecular weight present in 50 μg (left) and 25 μg (right) zymography are marked with red boxes.
Caption below figure 4 is incomplete or even confusing: “A) Ten micrograms of venom protein from C. aquilus (Ca), Crotalus polystictus (Cp) and C. molossus nigrescens (Cmn) were incubated for 2.5 h at 37° C”. Do you mean venom before electrophoresis, gel after electrophoresis or something else? Besides, obtained lines on zymogram have more differences than similarities. There is no point to indicate some specific differences where there is so many of them.
The correction was done (page 6, line 149). Also, boxes indicating differences between venoms were deleted due the high variability in SVMP.
How do you explain the lack of presence of SVMP described in literature (about 25 kDa)? Is it possible that you don’t see monomeric forms of SVMP in the conditions you run the gels?
Some SVMP are present as complexes, information was included (page 7, lines 184-185).
According to the obtained results that clearly indicate that not only metaloproteases were separated on gels with gelatin, I suggest the change of paragraph title. Metalloproteases Zymography suggests that SVSPs have not gelatinolytic activity and all observed results concern SVMPs. These results are very interesting and I do not deny their value but description in this case is a bit confusing.
The change of title was done (page 5, line 133), paragraph was reworded (page 5, lines 134-145 and the presence of SVSP with no gelatinolytic activity was discussed (page 6, lines 156-164).
Line 166 “In snake venoms, the most abundant components are SVMP and SVSP” this is non generally true. It may be for some viperid venoms, but there are also a lot of PLA2s. But it is absolutely not true for elapid venoms, where 3FTx toxins are the principal component.
Answer: the line was modified and corrected (page 7, line 182).
2.6 RP-HPLC
On the figure 8 (in the caption of course) it should be clearly marked that proteins in presented peaks were not identified. They are predicted on the base of literature data and there is no proof that mentioned proteins are actually present in those peaks.
The correction was done (page 9, lines 233-246 and page 10, lines page 251-253).
2.7 MALDI-TOF-MS
In my opinion this experiment does not have scientific value. There is no identification data for proteins in particular peaks. All information is a prediction on the base of literature data and from proteomic point of view they are illegitimate. How do you know that there are no multiple-charged ions on this mass spectrum? You reported that in analyzed venoms you observed proteins with higher mass than 50 kDa. Presented mass spectrum “ends” with 50 000 m/z (by the way it is not necessarily equal to mass) and most of enzymes analyzed in this paper have higher mass. Besides, while the mass of PLA2 observed on zymography agrees with m / z on mass spectra, in the case of SVMP the difference is huge. The zymography with gelatin indicate SVMPs with the mass of 80 kDa and more not about 25. In both paragraphs (2.3 and 2.7) you write that there are reports about SVMPs with lower mass than you observed. How can you explain this difference? Concluding, the mere comparison of mass spectra is interesting but it does not add anything. In my opinion these result does not have value and conclusions are not supported by facts.
The MALDI-TOF-MS assay was deleted.
3. Materials and Methods
Nowhere is there any information on how the venom was dissolved either for electrophoresis or for enzyme tests.
The information about how were dissolved the venoms is indicated in page 10, line 270.
4. Conclusions
“Some of the proteases found in the studied species showed molecular weights not previously reported…” these differences probably result from the fact that you observe multimeric forms of proteases. This subject should be discussed in this article.
Some SVMP could be present as complexes, this was discussed in page 7, lines 184-185.
“Our results show that Ca, Cp and Cmn venoms are mainly hemotoxic with the main groups of protease families usually found in Viperidae venom: metalloproteases, serine proteases, phospholipases A2, and hyaluronidases…” - neither phospholipases nor hyaluronidases are peptidases.
The correction was done (page 13, line 371).

Reviewer 3 Report
Abstract
1- line 20: Hemorrhaging - correct to bleeding
2- line 20-21: The authors claim that “each enzyme can have more than one enzymatic activity”. Please explain this phrase. I can not agree with this…
Introduction
1- line 39: The authors claim that "venom components have been poorly characterized”. I can not agree with this…The venom components are deeply studied however we do not know all the their actions.
2- line 54: correct to - Central Mexico endemic species.
Results and Discussion
2.1: Please rewritten this paragraph. The SDS-PAGE show very little proteins. This is not convencional/normal for snake venom. It was spectate more proteins in SDS-PAGE.
2.2: Please give points to the sentences. They are so long!
Zymography semens to be blurred. Please a new gel with more clear bands is necessary. Try zymography with different venom concentrations in order to be more clear.
Figure 3, 4, 6 and 7: Try to put symbols to express statistical differences. What the letters means? a, b, c, d…
Line 166: “In snake venoms, the most abundant components are SVMP and SVSP". This is not true for all genus and species. Please rewritten.
2.7 line 241: can be lectins molecular weights…. And about LAAO? If the venom is yellow there is LAAO!
3.2 line 279: venom dissolved in ??? Spin???
3.3.1, 3.3.2 and 3.3.3 venom samples (mg/mL ???) dissolved in ???? And so on…
Conclusion: The authors do not take a look at the LAAO presence. It is important to mention why…
Author Response
Reviewer 3
1- line 20: Hemorrhaging - correct to bleeding
The correction was done (page 1, line 20).
2- line 20-21: The authors claim that “each enzyme can have more than one enzymatic activity”. Please explain this phrase. I can not agree with this…
The sentence was corrected “Each enzyme can have more than one enzymatic activity” for “Each venom has several enzymatic activities” … (page 1, line 21).
Introduction
1- line 39: The authors claim that "venom components have been poorly characterized”. I can not agree with this…The venom components are deeply studied however we do not know all the their actions.
The correction was done (page 1, lines 39-40)
2- line 54: correct to - Central Mexico endemic species.
The correction was done (page 2, line 56).
Results and Discussion
2.1: Please rewritten this paragraph. The SDS-PAGE show very little proteins. This is not convencional/normal for snake venom. It was spectate more proteins in SDS-PAGE.
There are approximately 12 protein bands in SDS-PAGE gels; however each protein band could be more than one protein, also, the protein pattern banding of Cmn and Cp are according to previously reported by Borja et al. (2018) [23]and Mackessy et al. (2018) (page 2, lines 80-81).
2.2: Please give points to the sentences. They are so long!
The manuscript was revised for grammatical and topographical errors.
Zymography semens to be blurred. Please a new gel with more clear bands is necessary. Try zymography with different venom concentrations in order to be more clear.
The paragraph explaining the enzymatic activities of serine proteases was reworded and box indicating differences between the venoms was completed (page 3, lines 101-106). Also, the experiment using lower venom concentrations was repeated, however,the zymography resulted blurred and some trypsin like serine proteases of lower molecular mass were not detected, for this reason we kept the original Figure 2 in the manuscript (page 4, line 113).
Protein pattern by SDS-PAGE, and Serine proteases zymography for C. aquilus(Ca), C. polystictus(Cp) and C. molossus nigrescens (Cmn) venoms. (A) 10% SDS-PAGE and (B) Zymography. Trypsin-like zymography was performed using 50 μg (left) and 25 μg (right) of venom protein samples with BApNA substrate. After electrophoresis, according to methodology, a cellulose membrane with each serine protease was placed on top the gel and incubating for 2.5 h at 37 °C MWM Molecular weight markers (kDa). Proteases of low molecular weight present in 50 μg (left) and 25 μg (right) zymography are marked with red boxes.
Figure 3, 4, 6 and 7: Try to put symbols to express statistical differences. What the letters means? a, b, c, d…
Small letters indicate significant statistical differences; this is shown in each figure caption of figures 3, 4, 6 and 7. (page 5, line 132; page 6, line 155; page 8, lines 207-208 and page 9, lines 228-229)
Line 166: “In snake venoms, the most abundant components are SVMP and SVSP". This is not true for all genus and species. Please rewritten.
The correction was done (page 7, line 182).
2.7 line 241: can be lectins molecular weights…. And about LAAO? If the venom is yellow there is LAAO!
The possible association of yellow colour in the venoms with LAAO was discussed in page 2, lines 81-87.
3.2 line 279: venom dissolved in ??? Spin???
The information about how were dissolved the venoms is indicated in page 10, lines 270-272.
3.3.1, 3.3.2 and 3.3.3 venom samples (mg/mL ???) dissolved in ???? And so on…
Answer: the information about how were dissolved the venoms conditions and protein of venom for each assay is indicated in page 11 lines 284; 295, 302, and 309.
Conclusion: The authors do not take a look at the LAAO presence. It is important to mention why…
Possible presence of LAAO is discussed (page 2, lines 81-87).

Round 2
Reviewer 2 Report
Please make changes in 2.3 paragraph. Please clearly indicate which part of the text refers to figure 4A and which to 4B.
Line 161 zymography not zimography
Author Response
Open Review
(x) I would not like to sign my review report
( ) I would like to sign my review report
English language and style
( ) Extensive editing of English language and style required
( ) Moderate English changes required
( ) English language and style are fine/minor spell check required
(x) I don't feel qualified to judge about the English language and style
Yes | Can be improved | Must be improved | Not applicable | |
Does the introduction provide sufficient background and include all relevant references? | (x) | ( ) | ( ) | ( ) |
Is the research design appropriate? | (x) | ( ) | ( ) | ( ) |
Are the methods adequately described? | ( ) | ( ) | ( ) | ( ) |
Are the results clearly presented? | (x) | ( ) | ( ) | ( ) |
Are the conclusions supported by the results? | ( ) | (x) | ( ) | ( ) |
Comments and Suggestions for Authors
Please make changes in 2.3 paragraph. Please clearly indicate which part of the text refers to figure 4A and which to 4B.
Figures 4A and 4B were indicated in page 5, lines 135-136.
Line 161 zymography not zimography
The correction was done in page 6, line 161.
Submission Date
10 February 2019
Date of this review
30 Mar 2019 13:57:16
Reviewer 3 Report
Dear authors,
Figure 3, 4, 6 and 7: Try to put symbols to express statistical differences. What the letters means? a, b, c, d…
Small letters indicate significant statistical differences; this is shown in each figure caption of figures 3, 4, 6 and 7. (pa
This information is still incomprehensible. Please, provide the correct differences. If you would like to use letters we must explain what different letters means (A, B, C... different from x?). Who is being compared to whom?
Author Response
Rev 3
Open Review
(x) I would not like to sign my review report
( ) I would like to sign my review report
English language and style
( ) Extensive editing of English language and style required
( ) Moderate English changes required
(x) English language and style are fine/minor spell check required
( ) I don't feel qualified to judge about the English language and style
Comments and Suggestions for Authors
Dear authors,
Figure 3, 4, 6 and 7: Try to put symbols to express statistical differences. What the letters means? a, b, c, d…
Small letters indicate significant statistical differences; this is shown in each figure caption of figures 3, 4, 6 and 7. Figure 3 was improved.
This information is still incomprehensible. Please, provide the correct differences. If you would like to use letters we must explain what different letters means (A, B, C... different from x?). Who is being compared to whom?
Statistical differences were indicated using small letters (Tukey p<< span="">0.05); this is shown in each figure caption of figures 3, 4, 6 and 7. Figure 3 was improved.
Submission Date
10 February 2019
Date of this review
28 Mar 2019 16:03:46